# Simulation Models for Suicide Prevention: A Survey of the State-of-the-Art

Ryan Schuerkamp [1], Luke Liang [1], Ketra L. Rice [2] and Philippe J. Giabbanelli [1,*]

1   Department of Computer Science & Software Engineering, Miami University, Oxford, OH 45056, USA; schuerr2@miamioh.edu (R.S.); liangl5@miamioh.edu (L.L.)
2   National Center for Injury Prevention and Control, Centers for Disease Control and Prevention (CDC), Atlanta, GA 30341, USA
*   Correspondence: giabbapj@miamioh.edu

**Abstract:** Suicide is a leading cause of death and a global public health problem, representing more than one in every 100 deaths in 2019. Modeling and Simulation (M&S) is widely used to address public health problems, and numerous simulation models have investigated the complex, dependent, and dynamic risk factors contributing to suicide. However, no review has been dedicated to these models, which prevents modelers from effectively learning from each other and raises the risk of redundant efforts. To guide the development of future models, in this paper we perform the first scoping review of simulation models for suicide prevention. Examining ten articles, we focus on three practical questions. First, which interventions are supported by previous models? We found that four groups of models collectively support 53 interventions. We examined these interventions through the lens of global recommendations for suicide prevention, highlighting future areas for model development. Second, what are the obstacles preventing model application? We noted the absence of cost effectiveness in all models reviewed, meaning that certain simulated interventions may be infeasible. Moreover, we found that most models do not account for different effects of suicide prevention interventions across demographic groups. Third, how much confidence can we place in the models? We evaluated models according to four best practices for simulation, leading to nuanced findings that, despite their current limitations, the current simulation models are powerful tools for understanding the complexity of suicide and evaluating suicide prevention interventions.

**Keywords:** agent-based model; suicide prevention; system dynamics; microsimulation; network simulation; discrete event simulation

## 1. Introduction

Suicide is a leading cause of death and a serious public health problem worldwide [1,2]. Globally, more than 700,000 people died by suicide in 2019, representing more than one in every 100 deaths, with roughly 75% of deaths occurring in low-income and middle-income countries [2,3]. While low-income and middle-income countries have the highest rate of suicide deaths [2], suicide is one of the leading causes of death among young people in higher-income countries, such as the United States and Australia. In the United States, suicide is the second leading cause of death in people aged 10–14 and third leading cause in people aged 15–34 [1]. In Australia, suicide is the leading cause of death for people aged 15–44 [4]. Examining suicide rates globally shows that suicide has no age or racial and ethnic boundaries and is a serious global health concern. Because of the global prevalence and burden of suicide, the reduction of suicide fatalities has been prioritized by the World Health Organization as a global target and included as an indicator in the United Nations Sustainable Development Goals 2030 [2].

Studies show that suicide is a complex issue with a range of risk factors across all ages, sexes, and racial and ethnic groups, including mental health and behavioral disorders (e.g., conduct disorder) [5–9], chronic pain [10,11], perceived discrimination among racial

and ethnic minorities [12,13], alcohol and drug abuse [14,15], adverse childhood experiences [16,17], social isolation [18,19], unsafe media portrayal of suicidal behaviors [20–22], and access to lethal means among people at risk [23,24]. In analyzing risk factors related to suicide, it is critical to view factors through the lens of social ecology [25], understanding that suicidal behaviors reflect multiple factors, from the individual to the relationship, community, and societal levels. Because of these numerous and often dependent and dynamic factors, simulation models provide practical tools to help examine suicidal behaviors and outcomes and the effect of policy and program interventions, potentially yielding insights regarding those different policy and program interventions that offer the most promise for reducing suicide risk and ultimately preventing suicide fatality.

Modeling and Simulation (M&S) is an interdisciplinary research area that contributes to public health by combining expertise in computational social science [26], epidemiology, or health psychology. It has informed the development, implementation, and assessment of public policy for several decades [27,28] in areas including obesity [29–31], tobacco control [29], hospital scheduling and organization [32], and the COVID-19 pandemic [33–35]. M&S provides virtual laboratories to evaluate the effects of potential public health interventions, known as *what-if scenarios*. A model may account for resources, thereby providing a decision tool to help coordinate and allocate resources [27]. The process of building a model for suicide prevention serves to clarify relevant concepts [36] and their complex dynamics (e.g., feedback loops) and to identify areas where further data collection is needed [37].

As a result of the strong track record of M&S in addressing public health challenges, modelers have contributed to suicide prevention efforts on several occasions. Specifically, three types of models have been utilized. First, *conceptual* models such as concept maps and networks (i.e., graphs) that focus on identifying key concepts and relationships in the problem space instead of making point predictions (i.e., predicting future values under different scenarios) have been used to model the complex factors and relationships contributing to adolescent suicide [36,38,39], evaluate protective factors (e.g., social support) [40], and examine the role of anxiety and depression in suicidal thoughts [41]. Second, *mathematical* models (e.g., systems of differential equations) that represent the current system have investigated the spread of suicide in Greece [42] and predicted suicide rates in the United States, Brazil, and Sri Lanka [43]. Third, *simulation models* such as Agent-Based Models and System Dynamics have been applied to support regional suicide prevention planning in Australia [44], reduce youth suicide [45], and examine the potential effects of reducing access to firearms for those with drug and alcohol-related misdemeanors on firearm-related suicide deaths [46]. In summary, the three types of models are on a continuum (Figure 1). Every model starts as a conceptual model, as modeling teams need to identify relevant factors and their interactions. If a modeling project needs and uses numerical data (e.g., suicide rates), it becomes a mathematical model. If equations cannot be solved in closed form, the model needs to be simulated using a computer implementation, becoming a simulation model. In this paper, we focus on simulation models, as they provide actionable tools for evaluating the effects of potential interventions.

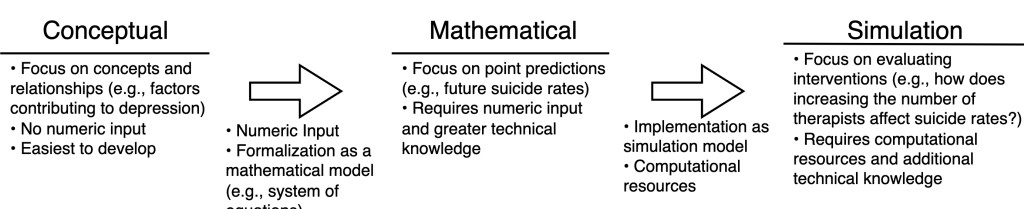

**Figure 1.** The continuum of conceptual, mathematical, and simulation models.

Our main contribution is to perform the first scoping review of simulation models for suicide. Our review is guided by three specific Research Questions (RQs):

1.  Which interventions are supported by previous models? Reviews of simulation models in public health occasionally show that groups of practitioners work in si-

los, possibly being unaware of existing tools [47]. As a result, several models may independently be developed to answer the same what-if scenarios, and none may be available for other scenarios of interest. Thus, a thorough inventory of the support offered by prior models can foster synergies across teams, provide a concrete toolbox for practitioners, and reveal areas in need of further efforts.

2. What are the obstacles preventing model application? The transition from simulation findings to the design and evaluation of interventions is often a difficult step [48–50]. Thus, identifying obstacles (particularly as they are shared across models) is an important way to ease this transition.

3. How much confidence can we place in models? Although a simulation model is ultimately an instrument [51], its intended users need to know the extent to which it can be trusted in a given application setting. Perfect trust does not exist, as a simulation is necessarily a simplification of reality; hence, the emphasis here is on knowing the limitations of a model and addressing them where possible.

The remainder of this paper is organized as follows. In Section 2, we briefly summarize the main *types* of simulation models encountered in this review. Then, we review the specific models by addressing each research question in turn in Sections 3–5. Finally, Section 6 discusses the current state of simulation models for suicide and provides suggestions for future work in this field.

## 2. Background

### 2.1. Agent-Based Models

Agent-Based Models (ABM) investigate how *individual behaviors and interactions* can produce system-level patterns and how individual differences affect the emergence of such patterns [52]. This approach has been used on many occasions to model complex health behaviors [53,54], and is one of the predominant systems science approaches in public health [55]. This approach consists of three main components: agents, interactions among agents, and interactions between agents and their environment. Agents (Figure 2) represent a person, firm, or entity capable of interacting with other agents and the environment [52,56]. First, the *agents' personal attributes* (e.g., age, income, and ethnicity) and behavioral rules affect their progression through the model and interactions with their environment and other agents. Agent-based modeling is a bottom-up process, as modelers define the agents and these interactions at the micro-level, then run simulations to observe the emergence of macro-level phenomena such as suicide clusters [57,58]. Agents may be equipped with goals (e.g., death by suicide), make decisions (e.g., plan to acquire and use a specific type of means of suicide), and learn behavior from their past experiences and other agents (e.g., one friend died by suicide using a specific means). Second, each agent has a *social network* detailing the nature (i.e., who has social ties with whom?) and function (i.e., what happens through each tie and when does it happen?) of interactions with other agents. For example, this allows the model to spread social norms or behaviors, such as when an agent is depressed and increases the odds of depression for other agents depending on the strength of their relationships. The structure of agents' social networks can change as the simulation progresses and as agents change states, behaviors, and locations. For example, an agent may move between locations, weakening or removing certain connections in its social network. Third, each agent has a *physical location* in the model, which can affect the agent's access to resources (e.g., mental health treatment) and states and may constrain their social networks.

Because Agent-Based Modeling is a very rich paradigm that allows complex socio-environmental phenomena to be captured, we provide brief and concrete examples of the agents' attributes, their interactions with others, and their relation to physical space. As is detailed in this review, two ABMs (collectively called the NYC ABMs) have been used to support interventions. They use several behaviors and attributes for agents, including substance use, psychiatric disorders, receiving mental health treatment, firearm carrying and ownership, violence, moving between neighborhoods, and suicidal behavior [46,59].

These models assign agents to communities that influence their social network and behavior. For example, certain agents may target potential victims of violence nearby. Violence can only occur if two agents directly interact in physical space, and proximal police officers can intervene and prevent the violence. Moreover, agents can interact with others in their social network, and experiences of violent victimization and incarceration influence the agent's risks of incarceration and homicide. Victims of violence are at greater risk of experiencing mental health disorders, inflicting self-harm, and being re-victimized.

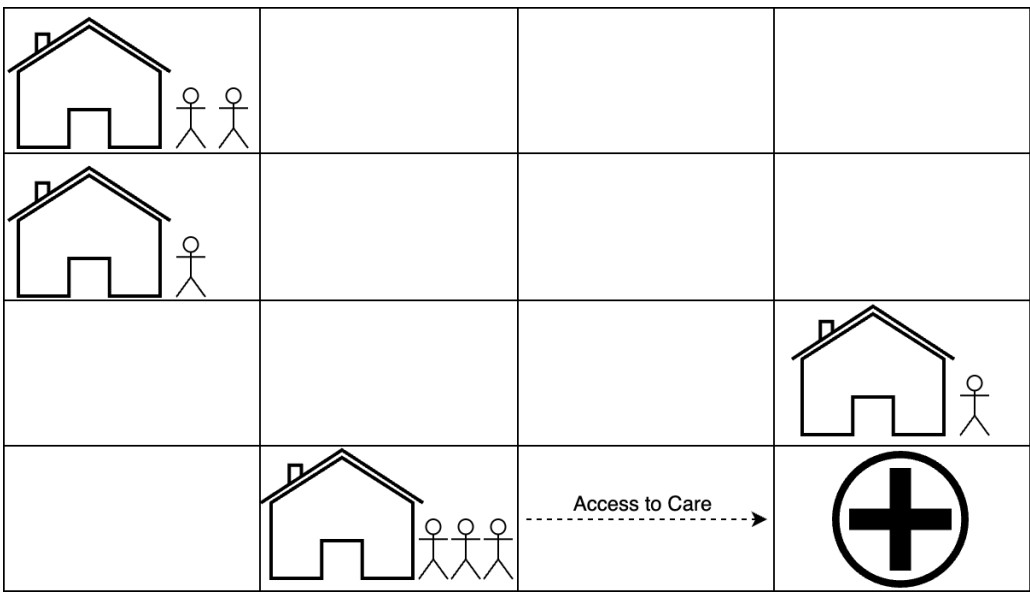

**Figure 2.** An example ABM depicting agents, their connections, and their access to care. Agents that are farther away have greater difficulty in accessing care.

It is important to note that the behavior of agents is not *separately* governed by agent–agent interactions and agent–environment interactions, as models have integrated how the agents' social networks can be *constrained* by physical considerations. Both ABMs mentioned above use physical location to create social networks in three steps. First, each agent is assigned to one of 59 communities in New York City. Second, the number of social connections for each agent is drawn from a uniform distribution ranging from zero to nine with a mean of four [59] or a range from one to nine with a mean of five [46]. Finally, agents are added to each other's social networks based on age, gender, race/ethnicity, education, firearm status (carrying and ownership), drinking status, and spatial proximity. Agents who are more similar and closer geographically are more likely to be in each other's social network. Social networks persist throughout the simulation and influence agent behavior, including prison sentences, homicides, and suicide risk probabilities.

The interactions between agents can be set throughout the model or can vary over time (i.e., static vs. dynamic); for the two NYC ABMs, social networks persist over the entire course of the simulation. Deciding which agents should share a social tie can involve empirical data (i.e., collected from the real-world), synthetic models (e.g., random graphs, scale-free or small-world network generators), or a mixture. The two NYC ABMs use a mixed process, assigning a uniform number of social ties to agents (synthetic) and then connecting them based on features obtained from datasets (empirical). Networks have properties, such as forming groups of individuals and being able to reach them indirectly via very few intermediaries. Formally, the joint presence of high clustering and low average distance results in labeling this network as small-world. While ABMs in health behaviors can explicitly rely on such properties, either when creating the network or when analyzing its empirical structure [60], the two NYC ABMs do not disclose any specific properties.

### 2.2. System Dynamics

Systems Dynamics (SD) is a simulation method that represents the problem domain as stocks (accumulated energy, items, and materials) and flows (rate of exchange between stocks) to investigate how causal interactions explain system behavior over time [56] (Figure 3). In contrast to ABM, modelers using an SD approach directly work with *system-level behavior* and connections between aggregate concepts (e.g., the effect of availability of mental health practitioners on suicide rates) [52]. Instead of representing the behavior of every individual, this approach focuses on the behavior of groups. For example, it can represent the population as a whole or break it down by gender. Interactions among groups can be represented, as can interactions between groups and the environment. Specifically, SD enables numerical analysis of a system of differential equations describing the flows between stocks. Although a system of differential equations is continuous, SD models are implemented using solvers that create approximate solutions via small discrete steps. SD models provide insight into feedback loops, nonlinear interactions, and delays in the system, enabling stakeholders to see the long-term system-wide outcomes of interventions [56]. These models have been frequently applied in public health research [61,62], in particular to explore the mechanisms of depression and possible treatments [63,64].

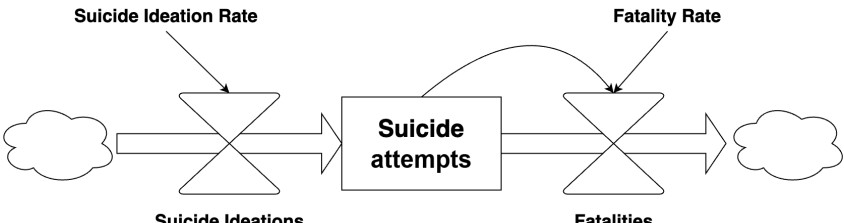

**Figure 3.** A simple System Dynamics model diagram for determining the number of suicidal ideations (i.e., suicidal thoughts).

### 2.3. Microsimulation, Network Simulation, and Discrete Event Simulation

In this subsection, we introduce three types of simulation models that were each used in a previous paper to model suicide. First, *microsimulations* model the effects of interventions on specific subpopulations [52], as illustrated in numerous works originating from the field of social and economic statistics [65,66]. Microsimulations are used when individuals have numerous characteristics that affect an outcome of interest and do not interact with one another. Unlike ABM, microsimulations focus on individuals, and while they can include spatial effects [67], they normally do not cover detailed interactions among individuals [52,68]. They use detailed datasets describing population characteristics, usually from national data sources such as the United States Census or Statistics Canada [65], to determine individual characteristics and identify which subpopulations are harmed or benefited by interventions.

Second, *network simulations* study the relationships (e.g., friendship, trust) between entities (e.g., people) and how these relationships can be impacted by the perceptions and behaviors of entities over time (Figure 4). This is a special case of ABM without location, and researchers may occasionally present a model as an ABM (a more familiar framework in the public health realm) even though it would more strictly qualify as a network model [69]. In a network simulation, each entity is a node and each relationship is an edge; these relationships can be directional (e.g., a person may considers someone else a friend even if the relationship is not mutual). The network can be quantitatively analyzed to identify influential entities and clusters (e.g., friend groups) and how they change over time. As a result, network simulations can investigate the structure and function of relationships and how they change in the system of interest. For instance, the structure of the network may be analyzed with respect to how it would affect the diffusion of a peer-driven intervention [70]. Note that networks have been abundantly studied over recent years with respect to suicide prevention, as summarized in the survey by Lopez-Castroman et al. [71]. Many these studies would be categorized as network analysis rather than simulations, as they look for

patterns in human-generated data. For instance, there may be a change in the language (e.g., increase in sadness) of social media users prior to a suicide attempt, which can be used by data mining algorithms exploiting social media platforms to detect possible suicide attempts. Another prominent example of network analysis in suicide research is the detection of suicide clusters from temporal social networks. As emphasized by recent research, "mechanisms underlying the contagion effect of suicidal thoughts and behavior are currently unclear" [72]. In contrast, network simulations create computer-generated data based on explicit mechanisms.

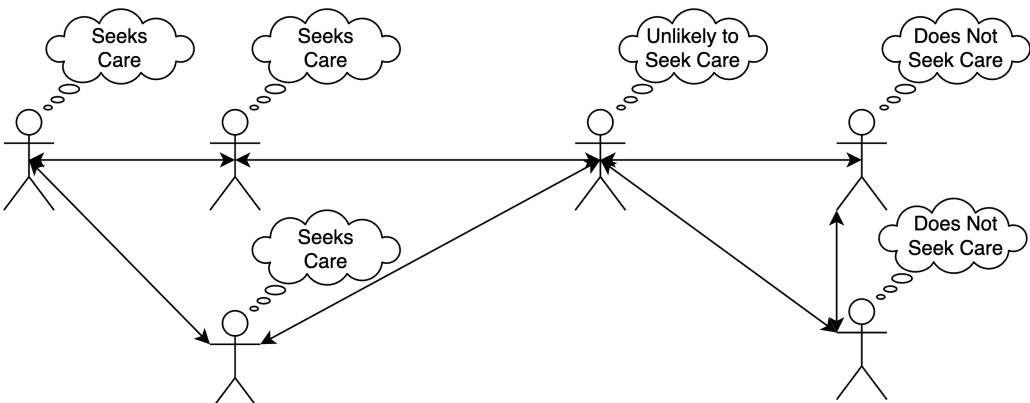

**Figure 4.** An example social network demonstrating the spread of care-seeking behavior.

Third, *Discrete Event Simulation* (DES) is used to model systems with queues for resources and processes with either a known duration [52] (for deterministic DES) or a variable one (for stochastic DES models). In DES, sequences of events such as arrival and departure are accounted for, and each event can start a future event (e.g., an arrival may produce the time for the next arrival). Modeled events in the context of suicide can include adverse childhood experiences such as violence or abuse [73]. As a result, discrete event simulation can help to evaluate how system constraints affect performance, and can identify bottlenecks and potential improvements. DES is often used in industrial engineering to manage resources. Applications include healthcare settings, such as sample studies on suicide screening in emergency departments [74].

## 3. RQ1: Which Interventions Are Supported by Previous Models?

We conducted a search with the key terms 'simulation' and 'suicide' on 14 February 2023, leveraging Google Scholar and selected results based on their relevance to computational modeling, identifying thirteen papers that provided simulation models regarding suicide. Among these thirteen articles, we found ten that supported interventions, two that leveraged ABMs to investigate "copycat" suicide [57,58], and one that used network simulation to investigate the effects of interpersonal loss on suicide [69]. In this section, we focus on the ten papers that implemented suicide prevention interventions, finding that 53 interventions were supported (Table 1). In examining the interventions modeled in each simulation, we matched them to the interventions identified in the World Health Organization (WHO) guide "Live life: an implementation guide for suicide prevention in countries" [75]. This resource provides a compilation of actions for suicide prevention and highlights four recommended interventions for global suicide prevention. Each intervention is a proposed action for countries to achieve the goal of preventing suicide. Using the WHO guide, we found four groups of papers with similar models and focus locations. For each group, we briefly examine the model(s) and interventions before analyzing and categorizing the collective set of 53 interventions. The four categories of suicide intervention are: (1) limiting access to the means of suicide, (2) interacting with the media for responsible reporting of suicide, (3) fostering socio-emotional life skills in adolescents, and (4) early identification, assessment, management and follow-up with respect to anyone affected

by suicidal behaviors (Tables 1 and 2). Note that we did not assign a suicide intervention category for the five interventions related to reducing unemployment and homelessness, increasing youth employment, or reducing domestic violence and childhood adversity, as the WHO does not have a category within their implementation guide for suicide prevention that relates to economic risk factors or risk factors associated with overall acts of violence. The WHO guide provides actionable direct suicide prevention strategies that low, middle, and high income countries can implement and monitor through community-level engagement. In addition, we did not assign a suicide intervention category for reducing psychiatric beds, as this is not considered an intervention, and may actually increase suicidal behaviors.

**Table 1.** Interventions supported by previous simulation models.

| | NYC ABM | | | Australia SD | | | | | Micro | DES |
|---|---|---|---|---|---|---|---|---|---|---|
| **Intervention** | **[59]** | **[46]** | **[76]** | **[77]** | **[45]** | **[78]** | **[44]** | **[79]** | **[80]** | **[74]** |
| Five-year firearm ownership disqualification for anyone with a psychiatric hospitalization | ✓ | | | | | | | | | |
| Five-year firearm ownership disqualification for anyone receiving psychiatric treatment | ✓ | | | | | | | | | |
| Firearms removed for five years after one alcohol-related misdemeanor conviction | | ✓ | | | | | | | | |
| Firearms removed for five years after one alcohol-related arrest | | ✓ | | | | | | | | |
| Firearms removed for five years after one drug-related misdemeanor conviction | | ✓ | | | | | | | | |
| Firearms removed for five years after one drug-related arrest | | ✓ | | | | | | | | |
| Firearms removed for 10 years after one alcohol-related misdemeanor conviction | | ✓ | | | | | | | | |
| Firearms removed for 10 years after one alcohol-related arrest | | ✓ | | | | | | | | |
| Firearms removed for 10 years after one drug-related misdemeanor conviction | | ✓ | | | | | | | | |
| Firearms removed for 10 years after one drug-related arrest | | ✓ | | | | | | | | |
| Firearms removed for five years after two or more alcohol-related misdemeanor convictions in five years | | ✓ | | | | | | | | |
| Firearms removed for five years after two or more alcohol-related arrest in five years | | ✓ | | | | | | | | |
| Firearms removed for five years after two or more drug-related misdemeanor convictions in five years | | ✓ | | | | | | | | |
| Firearms removed for five years after two or more drug-related arrests in five years | | ✓ | | | | | | | | |
| General practitioner training | | | ✓ | | ✓ | ✓ | ✓ | | | |
| Coordinated aftercare in those who have attempted suicide | | | | ✓ | | | | | | |
| School-based mental health literacy programs | | | ✓ | | | | | | | |
| Brief-contact interventions in hospital settings | | | ✓ | | | | | | | |
| Psychosocial treatment approaches | | | ✓ | | | | | | | |
| 20% reduction in the lethality of means | | | ✓ | | | | | | | |
| Reducing psychiatric beds | | | | ✓ | | | | | | |
| Increasing the capacity of community-based services | | | | | ✓ | ✓ | ✓ | ✓ | | |
| Post-attempt assertive aftercare | | | | | | ✓ | ✓ | ✓ | | ✓ |

**Table 1.** *Cont.*

| Intervention | NYC ABM | | | | Australia SD | | | | Micro | DES |
|---|---|---|---|---|---|---|---|---|---|---|
| | [59] | [46] | [76] | [77] | [45] | [78] | [44] | [79] | [80] | [74] |
| Social connectedness programs | | | | | ✓ | ✓ | ✓ | ✓ | | |
| Community-based acute care services | | | | | ✓ | ✓ | ✓ | | | |
| Technology-enabled crisis response | | | | | ✓ | | | | | |
| Technology-enabled coordinated care | | | | | ✓ | ✓ | ✓ | | | |
| Post-discharge peer support | | | | | ✓ | | | | | |
| Reducing childhood adversity by 20% or 50% | | | | | ✓ | | | | | |
| Increasing youth employment by 20% or 50% | | | | | ✓ | | | | | |
| Reducing total unemployment by 20% or 50% | | | | | ✓ | | | | | |
| Reducing domestic violence by 20% or 50% | | | | | ✓ | | | | | |
| Reducing homelessness by 20% or 50% | | | | | ✓ | | | | | |
| Community-based education programs | | | | | | ✓ | | | | |
| Family psychoeducation and support | | | | | | ✓ | ✓ | | | |
| Safety planning | | | | | | ✓ | ✓ | | | |
| Safe space services | | | | | | ✓ | ✓ | | | |
| General practitioner services capacity increase | | | | | | ✓ | ✓ | | | |
| Psychiatrist and allied health services capacity increase | | | | | | ✓ | ✓ | | | |
| Psychiatric hospital capacity increase | | | | | | ✓ | ✓ | | | |
| Awareness campaigns | | | | | | | ✓ | | | |
| Suicide helpline services | | | | | | | | ✓ | | |
| Community management of severe disorders | | | | | | | | ✓ | | |
| Mental health education programs | | | | | | | | ✓ | | |
| Services re-engagement programs | | | | | | | | ✓ | | |
| Online services | | | | | | | | ✓ | | |
| Hospital staff training | | | | | | | | ✓ | | |
| Services capacity increase | | | | | | | | ✓ | | |
| 12 week antidepressant treatment | | | | | | | | | ✓ | |
| 36 week antidepressant treatment | | | | | | | | | ✓ | |
| 52 week antidepressant treatment | | | | | | | | | ✓ | |
| Emergency department suicide risk screening for patients at least 10 years old | | | | | | | | | | ✓ |
| Hospital suicide risk screening for patients at least 12 years old | | | | | | | | | | ✓ |

**Table 2.** The number of interventions supported for each category defined by the WHO guide [75]. Note that we excluded six interventions: 'reducing psychiatric beds' was removed, and the five interventions related to increasing youth employment and reducing total unemployment, domestic violence, adversity, and homelessness do not fit into any of the WHO categories.

| Category | Interventions | % |
|---|---|---|
| Limit access to the means of suicide | $n = 15$ | 31.9% |
| Interact with the media for responsible reporting of suicide | $n = 1$ | 2.1% |
| Foster socio-emotional life skills in adolescents | $n = 5$ | 10.6% |
| Early identification, assessment, management and follow up of anyone who is affected by suicidal behaviours | $n = 26$ | 55.3% |
| Total | 47 | 100.0% |

The first group, on limiting access to means of suicide, consists of two papers that used ABMs to investigate the effects of fourteen interventions, implementing strategies to reduce access to firearms and investigating the effects on firearm-related homicide and

suicide deaths in New York City (NYC) [46,59]. The authors used ABMs to approximate 5% and 15% samples of the NYC adult population aged 18 to 64 years, discrete annual time steps with a 30-year time horizon to evaluate the long-term effects of interventions, and several national and local datasets to calibrate the model. The second group consists of six papers that used SD models to investigate the effects of 34 interventions on suicide rates in Australia [44,45,76–79]. The authors in this group frequently calibrated their model on historical data using national data for suicide deaths, and mainly selected interventions that had been the subject of a large-scale community trial at the time of publishing or that were the focus of government funding strategies. The remaining papers consisted of one model using microsimulation and one using DES. The microsimulation paper used a population of 300,000 adolescents and children with major depressive disorder (MDD) in the United States to investigate the effects of three interventions consisting of treatments for depression over 12, 36, and 52 weeks on suicide rates in adolescents and children with MDD [80]. The DES paper evaluated the effects of two interventions introducing suicide risk screening in the Children's National Hospital and their emergency department [74] by leveraging detailed hospital data.

Here, we have two observations to make after examining the collective set of 53 interventions. First, we observe that interventions can be concrete and actionable (e.g., five-year firearm ownership disqualification for anyone after a psychiatric hospitalization [59]) or more abstract (e.g., reducing childhood adversity by 20% or 50% [45]). The interventions considered by the Australian SD group tended to be more abstract than those considered by the other three groups (e.g., social connectedness programs [44,45,78]), allowing modelers and stakeholders to observe the effects of broader interventions that can be fulfilled by several different approaches. For example, social connectedness programs can encapsulate numerous different programs, such as after school programs for children and weekly events in a senior center. The interventions considered by the other three groups were more specific, providing stakeholders with a clear assessment of a precise action. Both broad and focused interventions are beneficial, and can support modelers in different ways. Broad interventions can assist modelers in identifying high impact areas, allowing more specific interventions to be designed, while simulating specific interventions can help stakeholders to select an optimal intervention out of a set of possibilities.

## 4. RQ2: What Are the Obstacles Preventing Model Application?

We identified two *limitations of interventions* considered by current models. First, although interventions are currently assessed by how much they reduce suicidal behavior, none of the 53 interventions across the ten reviewed papers accounted for *cost-effectiveness*, even though it is a critical metric. Indeed, the outcomes of an intervention (e.g., reduction in incidence, gain in quality-adjusted life years) should be related to its cost to ensure that limited resources can be best allocated. We note here that while a paucity of economic evaluations for suicide prevention evaluation was reported in the mid-2000's [81], several detailed analyses of cost-effectiveness have been undertaken within the last five years [82]. Thus, there may be a delay between these new analyses and the uptake of related practices in simulation modeling. In addition, it should be noted that certain existing models acknowledge resource constraints, and sets of three [78] or four to five [44] interventions have been evaluated instead of all possible interventions to reflect these constraints.

Second, certain considered interventions may be infeasible to implement. For example, reducing childhood adversity by 50% may be unrealistic, both because of the magnitude of change and because adversity would not be the sole target of any intervention. More specifically, as an intervention that reduces adversity may have other effects that affect outcomes, it is difficult to relate simulation outputs to a specific intervention. Additionally, certain interventions may be unsustainable. For example, the intervention of increasing psychiatric hospital capacity assumes a 50% increase in the annual rate of growth in public psychiatric hospital capacity (i.e., the maximum number of admissions per week), and this intervention is assessed over five-year and twenty-year time horizons [78], assuming a 50%

increase in annual growth over five or twenty years. Although the simulated interventions may not happen in the real world, they nonetheless provide valuable information by estimating the maximum potential effect. For instance, a simulation may show that even if a risk factor were to be massively lowered the impact on reducing death by suicide would be minimal; hence, any lower-level (or 'more practical') interventions would have low effectiveness as well.

In addition, we identified two *limitations within the data* used by the models. First, because models may have to include several datasets to cover different facets of suicide research [37], issues can arise when the combined datasets operate over *different time scales*. For example, modelers may aggregate the data operating at the more detailed time scale, create new points by extrapolation using the less detailed scale, or a mixture of these approaches. However, case studies in epidemic simulations provide words of caution, as "no aggregation or sampling policy tested was able to reliably reproduce results from the ground-truth full dynamic network" [83]. Second, there exists a *disconnect in time scales* between simulated data and real data. The simulation models (with the exception of System Dynamics) proceed in discrete time units or 'ticks'; for example, agents' behaviors are updated at one tick and then updated again at the next tick. In several cases, these ticks represent less than a day [44,45,79]. The behavior of the model is checked against real data to provide a comparison, thereby establishing the validity of the model. If a model has a simulation time tick of a month and the real data used for validation are provided on a yearly basis, then the endpoint of the simulation may be accurate (i.e., after twelve ticks) while the trajectory is erroneous. This creates issues for model reliability, particularly if the model is then used to support interventions that operate at a finer time scale than supported by the validation data. This challenge is not easy to resolve, as national surveys and surveillance datasets often report data yearly or biennially, whereas events involved in suicide attempts or death typically unfold over a much shorter time scale.

## 5. RQ3: How Much Confidence Can We Place in the Models?

While simulation models may produce point estimates that differ from reality, they are nonetheless powerful tools for evaluating interventions, especially if they follow best practices. For example, they can estimate how much suicide prevention interventions reduce suicide by, enabling comparison of interventions even if the point estimates do not closely align with reality. We evaluated each group of models according to four best simulation practices [84] (Table 3). First, we checked whether the model groups specified their time frames (i.e., the length of the simulation) and time granularity (i.e., the length of each tick/iteration). For example, the NYC ABM group used a time frame of thirty years and a time granularity of one year, meaning that predictions were produced each year for thirty years. Second, we examined whether each model group performed a sensitivity analysis to evaluate how much the model parameters affected the outputs. Third, we assessed whether the authors modeled heterogeneity in the population, such as differences in gender, age, and race/ethnicity. Fourth, we checked whether they used several data sources to inform their model. We found that all but the DES model fulfilled the four criteria; however, considering the DES model's goal of simulating hospital wait times for their specific hospital, modeling heterogeneity and leveraging several data sources is not very relevant in this case.

**Table 3.** Summary of whether the reviewed simulation model groups fulfilled four best practices.

| Model Group | Time Frame and Granularity | Sensitivity Analysis | Heterogeneity | Several Data Sources |
|---|---|---|---|---|
| NYC ABM | Y | Y | Y | Y |
| SD Australia | Y | Y | Y | Y |
| Micro | Y | Y | Y | Y |
| DES | Provided time frame but not granularity | Y | N/A | N/A |

Although the model groups fulfilled the criteria at a high level, there is room for improvement regarding the justification and alignment of the time frame with model users and the use of heterogeneity to analyze simulation outcomes. First, if the time frame and granularity are *not justified*, incorrect inferences may be drawn about the possible long-term benefits of an intervention [84]. For example, certain models disclosed their time frame without justifying it [44,45,79]. Second, the *time frame may not support the decision making* activities of its end users [84]. For example, the ABM NYC models used a time frame of thirty years to evaluate the long-term impacts of interventions around reducing firearm access. However, if its intended end users are policymakers, then they may need to evaluate the intervention and report its effects to constituents within a much shorter timespan. Finally, most models accounted for heterogeneity in their simulation and *not in their results*, raising the risk that an individual may only benefit when they are the assumed average person for the intervention, ignoring potential differences in effects across demographic groups [84]. For example, removing firearms from individuals experiencing suicidal ideation or a mental health crisis may lead to a larger decrease in male suicide deaths than female suicide deaths, as more males die using firearms than females [85]. Note that the need for improvement does not negate the current utility of these models. They can help us to understand the complexity of suicide and evaluate suicide prevention interventions, and can inform intervention design, selection, and implementation. Thus, making further improvements could increase the uptake of modeling in suicide prevention and broaden the use of these models.

## 6. Discussion

Suicide is a global public health challenge, and M&S has been widely used to address such challenges, including suicide prevention. M&S has helped identify the complex factors and relationships contributing to suicide, make point predictions of future suicide rates, and evaluate suicide prevention interventions. As a result of the strong potential of M&S to evaluate suicide prevention interventions, several simulation models have been developed, raising the risk of redundant efforts. Thus, a review of current simulation methods, the interventions they supported, and the obstacles hindering model application is needed to identify which additional interventions and features should be supported by future work.

The majority of the simulation models reviewed here focused on interventions that limit access to the means of suicide ($n = 15$) and interventions around early identification, assessment, management, and follow-up for people affected by suicidal behaviors ($n = 26$). In limiting access to the means of suicide, the primary approach was to reduce access to firearms for persons at risk of suicide. Those identified as being at risk in each simulation model were persons with a prior psychiatric hospitalization and persons with prior drug and alcohol-related arrests or convictions. While this approach limits access to firearms from those who have been institutionalized or incarcerated, it does not change access to firearms for those in the general population who may be at risk. For the strategy focused on early identification, assessment, management, and follow-up for those affected by suicidal behaviors, the approaches varied, including practitioner training, safety planning, coordinating care, providing clinical, psychosocial, and/or therapeutic approaches for care, increasing provider capacity, and planning for safety and follow-up for persons with a prior suicide attempt. These approaches recognize a need for increasing capacity among providers and communities in order to identify at-risk individuals and expand the types of care provided. Interventions underrepresented across all models included those focused on fostering social emotional life skills in adolescents, such as developing peer support programs, and interventions focused on interacting with the media for responsible reporting of suicide, such as developing safe messaging for suicide prevention.

Simulation models can play a pivotal role in suicide prevention efforts, as they are able to model both the independent and combined effects of multiple interventions. Broadening the scope of interventions to align with strategies based on the best available evidence can contribute to more practical simulation models that can directly inform public health

and help communities to focus resources on those prevention activities with the greatest potential to prevent suicide.

A model is a simplification of reality; hence, it is always subject to a set of limitations. An awareness of these limitations is important to avoid misuse of a model; hence, the present scoping review can help to set expectations for end users. In addition, because every individual model makes simplifications, we echo the conclusions of Lopez-Castroman that "multiple predictive models should be defined, implemented, tested, and combined in order to deal with the risk of suicidal behavior through an effective decision support system" [71]. Moreover, we emphasize that there are limitations that may not be easy to address. For example, a large gap between the time ticks of the model and the time granularity of the validation dataset is an important target for improvement; however, this represents a significant undertaking during data collection. In our review, we identified several limitations that can more readily be tackled through increased collaboration between modelers and model commissioners, stakeholders, or subject matter experts. In particular, the possible disconnect between the time horizon of a model and the needs of policy planners or evaluators is an item requiring particular attention. Ultimately, this scoping review is the first assessment of practices for simulation modeling in suicide, and our hope is that it can contribute to improvements models, modeling practices, and usage. Conducting such an assessment regularly would serve to further guide these efforts.

**Author Contributions:** Conceptualization, R.S., P.J.G. and K.L.R.; investigation, R.S. and P.J.G.; data curation, R.S.; writing—original draft preparation, R.S., L.L. and K.L.R.; writing—review and editing, P.J.G. and K.L.R.; visualization, R.S.; supervision, P.J.G.; project administration, P.J.G.; funding acquisition, P.J.G. and K.L.R. All authors have read and agreed to the published version of the manuscript.

**Funding:** P.J.G. was funded by an Intergovernmental Personnel Act (IPA) Mobility Program with the Centers for Disease Control and Prevention under IPA agreement #20IPA2009426-MOD2. R.S. and L.L. received funding from Miami University.

**Data Availability Statement:** No new data were created or analyzed in this study. Data sharing is not applicable to this article.

**Conflicts of Interest:** The authors declare no conflict of interest.

## Abbreviations

The following abbreviations are used in this manuscript:

| | |
|---|---|
| ABM | Agent-Based Models |
| DES | Discrete Event Simulation |
| M&S | Modeling and Simulation |
| MDD | Major Depressive Disorder |
| SD | System Dynamics |

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
