# Peer review of "Simulation Models for Suicide Prevention: A Survey of the State-of-the-Art"

_computers, doi:10.3390/computers12070132_

Round 1

Reviewer 1 Report

Much more attention should be paid to description of the models. That is, Section 2.1 should be expanded to be more specific about the models.

What behavioral rules are prescribed to individuals in the models and how there rules affect the inferences? What kind of decisions they make?

What kind of network is used (small world, scale-free or what?) in the models? Are individuals' network environments related to physical locations? etc

Author Response

We thank for the reviewers for the quick feedback and the very relevant technical questions. The questions and our answers are as follows.

Much more attention should be paid to description of the models. That is, Section 2.1 should be expanded to be more specific about the models.

>> Thank for the suggestion. We agree that Agent-Based Modeling is a very rich paradigm so there are many options as to how it can be applied to suicide, and readers would benefit from more specific descriptions. We expanded section 2.1 from 1 paragraph (previously) to 4 paragraphs (now).

What behavioral rules are prescribed to individuals in the models and how there rules affect the inferences? What kind of decisions they make?

>> Each group of model is highly customized, and models within a group can also have slightly different rules. To provide a clear picture to the readers about behavioral rules found in ABMs, we focused on one group of models and described it in section 2.1. We note that our research questions are primarily motivated by the application of modeling to suicide prevention, so we focused on interventions supported, obstacles to application, and confidence in the models. Detailing the models’ rules and inference processes are very interesting research questions as well, but they are more modeling- or technically-focused and hence would be difficult to put side-by-side with our application-focused research questions in a dedicated section.

What kind of network is used (small world, scale-free or what?) in the models? Are individuals' network environments related to physical locations? Etc

>> We appreciate this technical question. There were only 2 ABMs among the models that supported interventions, and they both had social networks. There are two other ABM models that examine copycat suicide, but they do not support interventions, and we do not discuss them in the paper. For the two relevant ABMs, we discussed their attributes and mechanisms in detailed in the expanded section 2.1. This includes the relation of networks to physical locations, the type of network constructed, and whether the authors disclosed any network property (they did not).

Reviewer 2 Report

The present work reviews a topic that directly targets mental health at the global level and implicitly the quality of life, namely a review of current simulation methods for suicide prevention, the interventions supported, and obstacles hindering model application needed to identify what additional interventions and features should be supported by future work. The article is well structured, it convinces the reader that the authors have done a responsible work of documentation and manages to demonstrate the fact that simulation models can play a pivotal role in suicide prevention efforts as they are able to model both the independent and combined effects of multiple interventions. I would like to make a suggestion to the authors, namely that the article should take into account the fact that some heuristic models can be included in the category of models that lend themselves to the topic of the current article, namely to name two: swarm models and pack of wolves models. Among the references I mention only J. Lopez-Castroman, B. Moulahi, J. Azé, S. Bringay, J. Deninotti, S. Guillaume, E. Baca-Garcia; Mining social networks to improve suicide prevention: a scoping review, J. Neurosci. Res. (2019); Riccardo Poli,  James Kennedy and Tim Blackwell;  Particle swarm optimization. An overview; Swarm Intel, DOI 10.1007/s11721-007-0002-0, (2007).

At the same time, the mention of deterministic discrete event system (DES) models made in lines 168 - 176 can be extended to the main DES models, respectively stochastic models and their applications in the subject of this article.

Author Response

We thank the reviewer for the prompt feedback and the valuable suggestions. The recommendations and our corresponding actions are detailed as follows.

The present work reviews a topic that directly targets mental health at the global level and implicitly the quality of life, namely a review of current simulation methods for suicide prevention, the interventions supported, and obstacles hindering model application needed to identify what additional interventions and features should be supported by future work. The article is well structured, it convinces the reader that the authors have done a responsible work of documentation and manages to demonstrate the fact that simulation models can play a pivotal role in suicide prevention efforts as they are able to model both the independent and combined effects of multiple interventions. I would like to make a suggestion to the authors, namely that the article should take into account the fact that some heuristic models can be included in the category of models that lend themselves to the topic of the current article, namely to name two: swarm models and pack of wolves models. Among the references I mention only J. Lopez-Castroman, B. Moulahi, J. Azé, S. Bringay, J. Deninotti, S. Guillaume, E. Baca-Garcia; Mining social networks to improve suicide prevention: a scoping review, J. Neurosci. Res. (2019); Riccardo Poli,  James Kennedy and Tim Blackwell;  Particle swarm optimization. An overview; Swarm Intel, DOI 10.1007/s11721-007-0002-0, (2007).

>> We are grateful for these pointers. We have enjoyed reading the paper on suicide prevention, which is certainly very closely related to our work. We have now cited it in multiple places. Specifically, we have doubled the description of network simulations under section 2.3, and we feature a shared conclusion in section 6. We have also keenly read the second reference, but its relation to suicide was not sufficiently clear for inclusion in our manuscript at present. We noted that the authors mentioned ‘suicide’ once, but it was solely in the context of an algorithm where “a particle kills itself when it is the worst in its neighborhood and generates a new copy of itself when it is the best”. We searched for additional papers on particle swarm optimization related to suicide prevention, but we only found studies on emotion recognition and we have already mentioned such algorithms in section 2.3. The same applies for grey wolf optimization, which was used in one publication (‘A Novel Optimizer Technique for Suicide Prediction In Twitter Environment’, 2021) that is also covered by the new content of section 2.3. We continue to welcome suggestions if there are any other publications on simulation models for suicide prevention.

At the same time, the mention of deterministic discrete event system (DES) models made in lines 168 - 176 can be extended to the main DES models, respectively stochastic models and their applications in the subject of this article.

>> This was a good idea. We have now added the terms deterministic and stochastic to our description of DES models.

Reviewer 3 Report

The main contribution of the paper is to perform the first scoping review of simulation models for suicide. The review is guided by three research questions: 1) Which interventions are supported by previous models? 2) What are the obstacles preventing model application? 3) How much confidence can we place in the models? The authors review simulation models by addressing each research question in turn, and discuss the current state of the models.

The paper is well written. I recommend publishing the paper in the MDPI Computers journal (Special Issue  Computational Modeling of Social Processes and Social Networks).

Author Response

We appreciate the kind and supportive words of the reviewer. 

Round 2

Reviewer 1 Report

I made some comments on the original version on the manuscript. The necessary changes have been made in the revised version. So now my recommendation is to publish the paper.